# Differences in the Quality Components of Wuyi Rock Tea and Huizhou Rock Tea

**DOI:** 10.3390/foods14010004

**Published:** 2024-12-24

**Authors:** Zhaobao Wu, Weiwen Liao, Hongbo Zhao, Zihao Qiu, Peng Zheng, Yuxuan Liu, Xinyuan Lin, Jiyuan Yao, Ansheng Li, Xindong Tan, Binmei Sun, Hui Meng, Shaoqun Liu

**Affiliations:** College of Horticulture, South China Agricultural University, Guangzhou 510642, China; wzb666@stu.scau.edu.cn (Z.W.); lww666@stu.scau.edu.cn (W.L.); zhao@scau.edu.cn (H.Z.); scau20222018004@stu.scau.edu.cn (Z.Q.); zhengp@scau.edu.cn (P.Z.); chegan09@stu.scau.edu.cn (Y.L.); imxyuanlin@stu.scau.edu.cn (X.L.); yaojuan@stu.scau.edu.cn (J.Y.); 17888679950@163.com (A.L.); txd@scau.edu.cn (X.T.); binmei@scau.edu.cn (B.S.); hui-meng@scau.edu.cn (H.M.)

**Keywords:** rock tea, HPLC, GC–MS, OAV, components, quality

## Abstract

Different origins and qualities can lead to differences in the taste and aroma of tea; however, the impacts of origin and quality on the taste and aroma characteristics of Wuyi rock tea and Huizhou rock tea have rarely been studied. In this study, high-performance liquid chromatography (HPLC), gas chromatography–mass spectrometry (GC–MS), and sensory evaluation methods were used to compare the quality components of Wuyi rock tea and Huizhou rock tea. The sensory evaluation showed that they each have their own characteristics, but the overall acceptability of Wuyi rock tea is ahead of Huizhou rock tea (*p* < 0.01). Biochemical experiments showed that HT was the highest in water leachables, about 43.12%; WT was the highest in tea polyphenols, about 14.91%; WR was the highest in free amino acids, about 3.38%; and the six rock teas had different health benefits. High-performance liquid chromatography showed that the theanine contents of WS and WR were 0.183% and 0.103%, respectively, which were much higher than those of other varieties. The OPLS-DA model predicted the factors that caused their different tastes, in order of contribution: CG > ECG > caffeine > EGCG > theanine. Ten volatile substances with OAV ≥ 1 and VIP > 1 were also found, indicating that they contributed greatly to the aroma characteristics, especially hexanoic acid, hexyl ester, and benzyl nitrile. The results of the correlation analysis showed that theanine was significantly correlated with taste (*p* < 0.05), and hexanoic acid, hexyl ester, and benzyl nitrile were significantly correlated with smell (*p* < 0.05). Substances such as theanine, hexanoic acid, hexyl ester, and benzyl nitrile give them their unique characteristics. Analysis of the differences in the quality components of the six rock teas can provide reference value for the cultivation and processing of rock teas.

## 1. Introduction

Tea is a healthy drink that is popular for its excellent health benefits and unique aroma and taste [1,2,3]. Wuyi rock tea is a traditional Chinese famous tea produced in the Wuyi Mountain area of northern Fujian. The tea trees grow in the cracks of the rocks and have a rock-like flavor (mellow and floral). Huizhou rock tea is based on Wuyi rock tea. The core production area is the Danxia landform mountains in eastern Guangdong. After professional research and development to improve the soil fertility, the species were introduced from Wuyi Mountain in Fujian and transplanted to Huizhou in Guangdong for cultivation. However, the different origins and processing methods result in a wide variety of tea flavors and aromas [4,5]. The climatic conditions, soil conditions, and processing methods of Wuyi Mountain and Huizhou are different, and the quality components are, therefore, not the same. The different quality components make them each have their own advantages and disadvantages, but there is a lack of scientific testing and comparison, and the factors affecting the differences in quality components also need to be studied urgently.

Rock tea is rich in caffeine and catechins, and less so in theanine, which largely determine the taste of the tea [6]. The three represent bitterness, astringency, and freshness, respectively [7,8]. It has been reported that the different proportions of these in the tea give Wuyi rock tea its different quality characteristics: Shuixian is mellow, soft, and sweet [8]; Tie Luohan is thick, fresh, smooth, delicate, and harmonious; and Rougui is mellow, fresh, smooth, and sweet. The water content, water leachables, soluble sugars, and free amino acids are directly related to the efficacy and taste of the tea. The water leachables reflect the rate of nutrient loss in the tea infusion and increase with storage time, which, together with soluble sugars and free amino acids, determines the health benefits of the tea. The higher the indicators of the three, the lower the rate of loss, the more nutrients that can be obtained, and the richer the nutritional value. Volatile substances directly determine the aroma of the tea infusion. 2-Ethyl-3,5-dimethylpyrazine, hexanoic acid, 3-hexenyl ester, (Z)-, hexanoic acid, hexyl ester, etc., are the main characteristic aroma components of Wuyi rock tea [9,10]. Research on Wuyi rock tea has been more comprehensive, but there is a lack of relevant research data on Huizhou rock tea.

High-performance liquid chromatography (HPLC) is widely used to detect soluble chemical components, and it can be used to accurately and efficiently detect substances such as caffeine, theanine, and catechins in tea [11]. The type and content of volatile substances in tea infusion are usually analyzed using gas chromatography–mass spectrometry (GC–MS) combined with a base peak chromatogram [12,13,14]. VIP > 1 and OAV ≥ 1 characteristic substances, whose types and contents cause aroma differentiation, were screened based on orthogonal partial least squares discriminant analysis (OPLS-DA), variable importance in projection (VIP), and odor activity value (OAV) [15,16]. This study comprehensively analyzes the taste and aroma characteristics of three varieties of tea from Huizhou and Wuyi using basic biochemical experiments, high-performance liquid chromatography, gas chromatography–mass spectrometry, and sensory evaluation analysis data. The quality components of the teas are compared scientifically, and the various factors that lead to their respective advantages and disadvantages are discussed and summarized, providing a theoretical basis for the cultivation environment and processing techniques of rock tea.

## 2. Materials and Methods

### 2.1. Samples

Wuyi Tie Luohan (WT), Wuyi Shuixian (WS), and Wuyi Rouqin (WR) are produced by Fujian Wuyi Mountain Ninety-nine Rock Tea Co. Ltd. (Fujian, China). The tea plants are 12 years old. All three types of rock tea are processed as follows: picking → sun-drying → withering (8~13 h) → fixing (350~380 °C, 10~11 min) → rolling (10~12 min) → water-removing roasting for 2 times (120~130 °C, 35~40 min) → charcoal roasting 2 times (110~125 °C, 12~14 h) → product (Figure 1). The tea samples were all produced between 2019 and 2023. Huizhou Tie Luohan (HT), Huizhou Shuixian (HS), and Huizhou Rougui (HR) were produced by Guangdong Echeng Rock Tea Co.(Guangdong, China). The tea trees are 12 years old. All three types of rock tea were made using the following process: picking → sun-drying → withering (10 h) → fixing (250 °C, 5 min) → rolling (5~8 min) → preliminary drying (135 °C, 1.5 h) → final drying (130 °C, 2 h) → roasting 4 times (135 °C, 1.5 h) → product (Figure 1). The tea samples were produced in 2022. The experiment was carried out in three biological replicates.

### 2.2. Chemicals

Reference standards, including ethyl caprate, theanine, catechins (C), epicatechins (EC), gallocatechins (GC), epigallocatechins (EGC), epicatechin gallate (ECG), gallocatechin gallate (GCG), and epigallocatechin gallate (EGCG), were purchased from Shanghai Yuanye Biotechnology Co., Ltd. (Shanghai, China). Caffeine standard was purchased from Beijing Weiye Metrology Technology Research Institute (Beijing, China). Alkane standard solutions (C8–C40) for calculating the linear retention index (RI) were provided by TanMo Quality Testing Technology Co., Ltd. (Beijing, China). The internal standard solution was prepared in dichloromethane before use [17]. Ultrapure water was prepared using a Barnstead GenPure Pro system (Thermo Fisher Scientific, Waltham, MA, USA).

### 2.3. Biochemical Experimental Analysis of Water Content, Water Leachables, Soluble Sugars, Free Amino Acids, and Tea Polyphenols

The determination of moisture content, soluble sugars, free amino acids, and tea polyphenols was carried out in accordance with national standards GB/T 8304-2013, GB/T 8305-2013, GB/T 8314-2013 and GB/T 8313-2018 [18,19,20,21,22], and the water leachables were determined using anthrone colorimetry. The concentrations of the three indicators were calculated using a standard curve (Appendix A).

### 2.4. High-Performance Liquid Chromatography Analysis of Caffeine, Theanine, and Catechins

To determine the content of caffeine, theanine, and catechin, high-performance liquid chromatography (Waters Alliance 2695, 2489 UV/Vis; Waters Technologies, Milford, MA, USA) was used with a 2489 ultraviolet/visible detector (Waters Technologies, Milford, MA, USA), and the retention times were compared with those of the standard substances. A calibration curve was used for quantitative analysis [23]. The concentrations of caffeine, theanine, and catechins were calculated using a standard curve (Appendix A).

For the extraction of caffeine, 0.10 g of tea powder and 0.45 g of magnesium oxide were added to a 50 mL centrifuge tube, and the mixture was soaked with 30 mL of 100 °C ultrapure water for 30 min. The reaction system was extracted by ultrasonic bath shaking for 30 min. Afterwards, the mixture was centrifuged at 10,000 rpm for 3 min, and the supernatant was filtered through a 0.22 µm Millipore membrane (Jinteng Experimental Equipment Co., Ltd., Tianjin, China). Then, 10 µL of the filtrate was injected onto an XSelect HSS C18 SB column (4.6 × 250 mm, 5 mm, Waters Technologies, Milford, MA, USA) at a column temperature of 35 ± 1 °C and a flow rate of 0.9 mL/min. The mobile phase was 100% methanol (A) and 100% ultrapure water (B). The detection wavelength for caffeine was set to 280 nm. The analysis was performed in triplicate.

For the extraction of theanine, 0.10 g of tea powder was placed in a 10 mL centrifuge tube and soaked with 10 mL of 100 °C ultrapure water for 30 min. After that, the mixture was centrifuged at 6000 rpm for 10 min, and the supernatant was filtered through a 0.22 μm Millipore membrane (Jinteng Experimental Equipment Co., Ltd., Tianjin, China). Then, 10 μL of the filtrate was injected into an RP-C18 column (250 mm × 4.0 mm, 5 μm, Waters Technologies, Milford, MA, USA) at a column temperature of 35 ± 1 °C and a flow rate of 0.5 mL/min. The mobile phase was 100% ultrapure water (A) and 100% acetonitrile (B). For the specific HPLC operating procedures, please refer to Mei et al. [24]. The detection wavelength for theanine was set to 210 nm. The analytical results were obtained in triplicate.

For the extraction of catechins, 0.20 g of tea powder was placed in a 10 mL centrifuge tube, 8 mL of 70% methanol (prepared with ultrapure water) was added, and the reaction system was mixed and extracted in an ultrasonic water bath for 30 min. After that, it was centrifuged at 10,000 rpm for 3 min, and the supernatant was filtered through a 0.22 μm Millipore membrane (Jinteng Experimental Equipment Co., Ltd., Tianjin, China). Then, 10 μL of the filtrate was injected into an XSelect HSS C18 SB column (4.6 × 250 mm, 5 mm, Waters Technologies, Milford, MA, USA). A gradient elution method was used, with the mobile phases being 0.1% formic acid in water (A) and 100% acetonitrile (B). The gradient elution started at 8% B and lasted for 5 min, increased from 5 to 14 min to 25%, and decreased from 14 to 30 min to 8% [24]. The detection wavelength for catechins was set to 280 nm. The analysis was performed in triplicate.

### 2.5. Gas Chromatography–Mass Spectrometry Analysis of Volatile Compounds

Volatile compounds were extracted and analyzed using headspace solid-phase microextraction/gas chromatography–mass spectrometry (HS-SPME/GC–MS) [17]. The analysis was performed using an Agilent 7890B gas chromatograph coupled to an Agilent 5977A mass spectrometer (Agilent, Santa Clara, CA, USA) and equipped with an HP-5MS capillary column (30 m × 0.25 mm × 0.25 μm film thickness). In short, 2.0 g of tea powder, 5.0 mL of saturated sodium chloride solution (prepared with ultrapure water), and 0.0864 g of ethyl caprate were added to a 40 mL headspace bottle, quickly sealed, and then preheated at 80 °C for 15 min. After that, solid-phase microextraction (SPME) was performed at 80 °C for 40 min using a divinylbenzene/Carboxen/polydimethylsiloxane (DVB/CAR/PDMS) fiber (inner diameter 50/30 μm, length 2 cm; Supelco, Darmstadt, Germany). After the extraction, the SPME fiber was placed in a gas chromatograph–mass spectrometer at 250 °C for 3 min.

The carrier gas was high-purity helium (purity ≥ 99.99%), the column flow rate was set to 1.0 mL/min, and the solvent delay time was 4 min. The initial temperature was 50 °C for 1 min, then the temperature was increased at a rate of 5 °C/min to 220 °C for 5 min. The electron energy and temperature of the ion source were 70 eV and 230 °C, respectively. The scanning range was 30–400 amu. The analysis was performed in triplicate.

The concentrations of volatile compounds were quantified using an established method. In brief, the ratio of the peak areas of the internal standard and the target compound was calculated, and the ratio of the two values reflected the actual concentration of the target compound. In addition, the substances were identified by searching the mass spectrometry database of the National Institute of Standards and Technology (NIST, https://webbook.nist.gov/, accessed on 10 October 2024) and using the retention indices (RI) of the n-alkanes C9 to C21 to determine the compounds. The RI is calculated as follows:RI = 100n + 100 [t_R_(x) − t_R_(n)]/[t_R_ (n + 1) − t_R_(n)]
where t_R_(x) is the retention time of compound x, and t_R_(n) and t_R_(n + 1) are the retention times of the alkanes with carbon atoms numbered n and n + 1, which elute immediately before and after the compound. A match is considered acceptable when the calculated RI is similar to the standard RI or when the mass spectrum match factor is greater than 90.

### 2.6. Calculation of the Odor Activity Value (OAV)

OAV is the ratio of the concentration of a volatile substance in an aqueous solution to its odor threshold. It is commonly used to assess the degree to which volatile compounds contribute to a characteristic aroma [25]. The formula is
OAV = C/OT
where C is the concentration of the compound (μg/kg) and OT is the compound’s threshold in water (μg/kg).

### 2.7. Sensory Evaluation

The overall acceptability of the tea infusion was scored by a tea expert from South China Agricultural University who is qualified as a national senior tea taster. A total of 20 judges participated, and all judges have received professional training in tea evaluation and have more than five years of experience in sensory descriptive analysis of tea. The evaluation criteria were based on the “Standard Method for Sensory Evaluation of Tea” (GB/T 23776-2018) [26]. First, 5 g of each tea sample was placed in a covered bowl and 110 mL of boiling water was added to brew the tea (the first brew was smelled after 1 min and tasted after 2 min; the second brew was smelled after 1 min and tasted after 3 min; the third brew was smelled after 1 min and tasted after 5 min). Each judge rated the taste and aroma. The average score was calculated using a maximum score of 10 (Appendix A).

### 2.8. Statistical Analysis

The three replicates of the raw data were processed using Microsoft Excel 2021. To analyze significant differences between the three samples, a one-way analysis of variance (ANOVA) and Tukey’s post hoc test were performed using the SPSS 27 software package (SPSS Inc., Chicago, IL, USA), and bar charts were plotted based on the analysis results. Bar charts, stacked charts, radar charts, and heat maps were created using Origin 2024 statistical software (OriginLab Corporation, Northampton, MA, USA). Orthogonal partial least squares discriminant analysis (OPLS-DA) and variable importance in projection (VIP) were performed using SIMCA (Version 14.1, Umetrics, Umea, Sweden) software. TBtools (Version 2.136, Guangzhou, China) was also used to plot the heatmaps.

## 3. Results

### 3.1. Quantitative Analysis of Water Content, Water Leachables, Soluble Sugars, and Free Amino Acids in Rock Tea

The water content, water leachables, soluble sugars, and free amino acids of rock tea were quantitatively analyzed using basic biochemical experimental analysis (Appendix A). The results showed that the water content of the six types of rock tea varied greatly, with extremely significant differences between the different origins of Rougui and Shuixian; the water leachables content of Tie Luohan varied extremely significantly, with HT being about 7% higher than WT; in terms of soluble sugar content, WR was the highest, while WS was the lowest at 3.32%, but there were no significant differences between the same varieties; and in the content of free amino acids, WR had the greatest advantage, and there were very significant differences between the same varieties from different origins; among the three groups of rock tea, there were also significant differences in the content of tea polyphenols. Free amino acids and tea polyphenols contribute significantly to the taste of tea, so it is speculated that they may be the main factors determining the taste of rock tea from different regions. The phenol-to-amino acid ratio was calculated based on free amino acids and tea polyphenols. When the levels of tea polyphenols and free amino acids reach a certain level, a lower phenol-to-amino acid ratio can obtain a fresh and mellow taste [27]. The results show that there are extremely significant regional differences between the three types of rock tea (Figure 2).

### 3.2. Quantitative Analysis of Caffeine, Catechins, and Theanine in Rock Tea

Caffeine, catechins, and theanine of rock tea were quantified using high-performance liquid chromatography (Appendix A). For Shuixian, theanine, C, and GCG were all highly significant lower in HS than in WS; while GA, EGC, and EGCG were all significantly higher in HS than in WS, but the difference in caffeine was not significant. For Rougui, HR’s theanine, C, and GCG were all significantly lower than WR; while HR’s caffeine, GA, and EGCG were all extremely significantly higher than WR. For Tieguanyin, the GA and ECG of HT were both extremely significantly lower than WT, and the caffeine was significantly lower than WT; while the GC, EGC, C, EC, GCG, and CG of HT were all extremely significantly higher than WT, and the EGCG was significantly higher than HT; however, the theanine content did not differ significantly. Overall, the total catechin content of HT was extremely significantly higher than that of WT. Significant differences were found among the 11 compounds tested in tea samples from the same variety but different origins, and it is speculated that they may be the factors responsible for the taste difference between Wuyi rock tea and Huizhou rock tea (Figure 3).

OPLS-DA is a multivariate statistical analysis method that can amplify group differences and reduce within-group differences, and it can be used to visually simplify HPLC results [28,29,30]. The score plot shows that there are significant differences in the non-volatile characteristics of the three groups of rock tea, and the three parallel repeats are good (Figure 4A). The cross-validation results show that the OPLS-DA discriminant model is not overfitted and the model is relatively reliable (Figure 4B). VIP values are used to evaluate the degree to which each variable contributes to the model. Variables with VIP > 1 contribute significantly to the model [31]. The results show that the VIP values of CG, ECG, caffeine, and EGCG are greater than 1, and they are most likely the factors for the different tastes of the three groups of rock tea (Figure 4C). The VIP values of GCG, EGC, GC, C, and theanine are close to 1, and they may also be potential factors determining the taste (Appendix A).

### 3.3. Volatile Components of Rock Tea

#### 3.3.1. Identification and Analysis of Volatile Substances

A total of 148 volatile substances were detected in the six rock teas using gas chromatography–mass spectrometry (GC–MS), including 65 in WR, 58 in HR, 57 in WT, 52 in HT, 51 in WS, and 49 in HS (Appendix A). On this basis, we counted the types of aroma substances in the six rock teas and found that the types of ester substances were the most numerous, followed by alkenes and alcohols, which suggests that the diversification of esters, alkenes, and alcohols may be the reason for the aroma differences in the six rock teas (Figure 5A). In order to further scientifically explore this difference, we selected the volatile substances common to the six rock teas and calculated their contents. The results showed that there were 23 aroma substances in the six rock teas (Figure 5B). Among these, there were seven esters, five alcohols, four ketones, two aldehydes, two alkenes and three other substances (Appendix A) [32,33,34,35,36,37,38,39].

We used cluster analysis to create a heat map and found that the volatile substances in the four rock teas WR, WS, HR, and HT were mostly upregulated (Figure 5C). For Rougui, WR showed a more obvious upregulation of benzyl nitrile, hexanoic acid, hexyl ester, hexyl benzoate, hexanoic acid, 3-hexenyl ester, and (Z)-, while HR showed upregulation of methyl salicylate, methyl palmitate, and neodiospyrin, which can produce a fresh fruity or fatty aroma. For Shuixian, (E)-hex-2-enyl acetate, benzyl alcohol, and α-ionone were significantly upregulated in WS, while limonene, methyl heptinone, and furfural were upregulated in HS. The aroma substances can produce floral and fruity aromas in Shuixian, and HS also has a burnt aroma. For the Tie Luohan, only phytone and phytol were significantly upregulated in the WT, which can produce floral and fatty aromas. The aroma components in the HT are more abundant, with abundant linalool, methyl salicylate, geranyl propionate, 1,2,3,4-tetrahydro-1,1,6-trimethylnaphthalene, and methyl heptinone, which can produce a fresh floral and fruity aroma [34,35,36,37].

#### 3.3.2. Screening for Volatile Substances

The threshold (OT) is the lowest concentration at which a compound can be perceived [40]. The odor activity value (OAV) is the ratio of the concentration of a compound to its threshold, which indicates the degree to which the compound contributes to the aroma [9]. The VIP value reflects the contribution of a variable to the overall model fit and classification ability. Variables with a VIP value greater than 1 are considered particularly important to the model [41]. In order to explore the characteristic volatile components of the six types of rock tea, an OPLS-DA model was constructed based on the 23 identified common volatile compounds (Figure 6A). The results showed that the prediction was relatively reliable and there was no overfitting (Figure 6B). The VIP score plot revealed that among the 23 common volatile compounds, there were 10 important volatile compounds with VIP values greater than 1, including 4 esters, 2 alcohols, 2 aldehydes, 1 alkene, and 1 other compound (Figure 6C).

Volatiles with VIP > 1 and OAV ≥ 1 were selected as potential characteristic volatiles (Table 1). The five substances 2,4-heptadienal, (E)-hex-2-enyl hexanoate, hexanoic acid, hexyl ester, (Z)-hex-3-enyl hexanoate, and linalool provide the fruity–floral aroma, while benzyl nitrile provides the spicy aroma and methyl salicylate provides the fresh aroma [34,37]. For Rougui, the higher total amount of fruity-floral substances and the high OAV value of WR make the fruity–floral aroma more intense, while the HR has a higher content of methyl salicylate and neo-geraniol, which have a fresh odor profile, and a high OAV value, which can increase the difference between the two odors (Figure 6D). For Shuixian, WS has a high content of (E)-hex-2-en-1-ol, which is fruity–floral [33], while HS has a high content of furfural, which often has a fire-like aroma [35]. For Tie Luohan, the WT has a low content of volatile substances, while the HT has a higher content of methyl salicylate and linalool, which provide a fresher floral aroma [34]. Taking into account the OAV and VIP values, the differences in the contents of the five substances 2,4-heptadienal, (E)-hex-2-en-1-ol, hexanoic acid, hexyl ester, methyl salicylate, and benzyl nitrile are likely to be the main reason for the differences in the aromas of the six rock teas (Table 2).

### 3.4. Sensory Evaluation of Rock Tea

According to expert assessment, in terms of overall acceptability, Wuyi rock tea (WT, WR, WS) is extremely significantly superior to Huizhou rock tea (HT, HR, HS) (Figure 7B). In terms of taste, HS has the strongest bitter and astringent flavors and the lowest freshness and fullness, while WS has the lightest bitter and astringent flavors and a higher level of freshness and fullness; HR has significantly higher bitter and astringent flavors, and a level of freshness and fullness slightly higher than the lowest value, while WR has a very light astringency and a high freshness and fullness; HT has a relatively high astringency and a relatively low freshness and fullness; and WT has a balanced astringency and a freshness and fullness that is between the upper and lower values (Figure 7A).

In terms of smell, the milk fragrance of the six rock teas is relatively weak, the honey fragrance is slightly stronger, and there is no significant difference between them. For Rougui, the roasted fragrance of HR is much heavier than that of WR, and the flower and fruit fragrance score is much lower than that of WR. For Shuixian, the roasted fragrance of HS is also much higher than that of WS, and the flower and fruit fragrance is lighter, lower than that of WS. For Tie Luohan, the roasted fragrance scores of the two rock teas are similar, and the flower and fruit fragrance of HT is slightly than WT (Figure 7A). Both Huizhou rock tea and WT have a strong roasted fragrance; and for the floral and fruity fragrance, WR is the strongest, followed by WS and WT. There is a big difference between these three, and they are all higher than Huizhou rock tea. The remaining three Huizhou rock teas have a weak floral and fruity fragrance (Appendix A).

### 3.5. Analysis of the Correlation Between the Flavor and Aroma Characteristics of Rock Tea and the Results of the Sensory Evaluation

Taste and aroma are the two most important factors in evaluating tea [42,43]. To further explore the correlation between the taste and aroma of rock tea and sensory evaluation, we conducted a correlation analysis of the 11 main non-volatile substances and 9 key volatile substances in 6 types of rock tea with the results of the sensory evaluation. The heat map shows that only theanine has a significant relationship with taste. Theanine is extremely significantly positively correlated with mellow and significantly negatively correlated with bitter (Figure 8A); in terms of aroma, hexanoic acid, hexyl ester, and benzyl nitrile are significantly positively correlated with floral, fruity, and honey aromas, while significantly negatively correlated with roasted aroma (Figure 8B). Methyl salicylate is significantly negatively correlated with frankincense. The remaining substances are not significantly correlated with sensory evaluation (Appendix A).

## 4. Discussion

### 4.1. Theanine Gives Rock Tea Its Unique Taste

Taste is an important part of tea evaluation, accounting for about 35% of the total score [44]. Among them, theanine is a key component that determines the freshness and richness of the tea infusion. Theanine content is related to the processing technology, and theanine content fluctuates during the processing. On the one hand, under the action of high temperature, theanine degrades faster, and on the other hand, protein hydrolysis occurs, and theanine gradually accumulates, but, overall, theanine tends to decrease [45]. In addition, the place of origin of the tea may also lead to differences in theanine content [46]. The OPLS-DA model shows that there are significant differences in the taste of the same variety of rock tea. The tea infusion sensory evaluation team also believes that there are very significant differences in the same variety of rock tea, and that rock tea from Wuyi Mountain is superior to rock tea from Huizhou. The taste of the tea infusion is closely related to flavor substances, and theanine has been identified as a characteristic and differentiated flavor substance of rock tea.

It is worth exploring that in this experiment, there was no significant relationship between theanine and freshness, replaced by a very significant positive correlation with mellowness and a significant negative correlation with bitterness. The richness of the tea infusion refers to the fact that the tea infusion is rich in substances such as tea polyphenols, amino acids, and caffeine, and gives a strong, heavy mouthfeel. Theanine can undergo a Maillard reaction with reducing sugars during processing to produce many flavor substances, which are closely related to richness. Feng et al. found a special flavor enhancer, N-(1-carboxyethyl)-6-(hydroxymethyl) pyridinium-3-ol (alapyridaine), and proposed the new insight that alapyridaine contributes to the umami and sweetness of Wuyi rock tea [47]. A growing number of studies have shown that theanine has a masking effect on the bitter taste of tea infusion.

We found that theanine content in the six rock teas follows a pattern: Wuyi rock tea generally has much higher theanine content than Huizhou rock tea, and Huizhou rock tea has very low theanine content, which coincides with the evaluation results. For Shuixian and Rougui, the theanine content of Wuyi rock tea is significantly higher than that of Huizhou rock tea, and it is also more mellow and has less bitterness. For Tie Luohan, there was no significant difference in theanine content between the two rock teas, and even though Wuyi rock tea contains slightly more theanine, the evaluation results did not show a significant difference between them.

Combining the processing techniques of the two rock teas, Huizhou rock tea is dried at temperatures of 150 °C and 130 °C, while Wuyi rock tea is carbon-roasted at lower temperatures multiple times. It can be speculated that high temperatures may lead to the low theanine content of Huizhou rock tea. Lin et al. showed that the Shuixian from Wuyi with charcoal roasting retained the most amino acids and scored the highest in sensory evaluation, and had the best rock tea quality [48]. Zhang et al. showed that theanine content can be used as an indicator of the difference between lower-grade teas with different degrees of roasting, and that the higher the degree of roasting, the lower the theanine content [49]. In addition, it is possible that the different growing conditions of the tea trees in the two places have resulted in lower theanine content.

Therefore, if we want to ensure the successful introduction and transplantation of Wuyi rock tea in Huizhou, it is extremely important to experiment with more suitable processing techniques, such as charcoal roasting or controlling the roasting temperature.

### 4.2. Hexanoic Acid, Hexyl Ester, Benzyl Nitrile, Methyl Salicylate, and 2,4-Heptadienal, (E,E)-Form the Characteristic Aroma of Rock Tea

Aroma is an important component in the evaluation of tea, accounting for about 30% of the total score [44]. The unique proportion of volatile substances in tea gives people different olfactory experiences and is a key factor in distinguishing different teas [50]. In this study, the volatile substances in six rock teas were detected by gas chromatography–mass spectrometry (GC–MS). They all have different types and contents of substances, especially 23 common components, which make them each have their own advantages and disadvantages.

Substances with VIP > 1 and OAV ≥ 1 are important, and a correlation analysis can be used to further explore the volatile substances that determine the aroma of tea. Hexanoic acid, hexyl ester has a floral and fruity aroma. It has a high OAV value in WR and WS, but a low OAV value in HR and HS. The evaluation results also show that the floral and fruity aroma and honey aroma of these two Wuyi rock teas are stronger, and the roasted aroma is weaker. Therefore, we can conclude that hexanoic acid, hexyl ester contributes greatly to the floral and fruity characteristics of WR and WS, while greatly weakening the roasted aroma produced during the roasting process, which is consistent with the research of Xu et al. [51]. Similarly, Wuyi rock tea has a higher content of benzyl nitrile, which can provide a spicy aroma, form a rich floral and fruity aroma, and mask the roasted aroma produced during processing. Benzyl nitrile is also a marker for Qin et al. to distinguish the Rougui-scented Fenghuang Dancong [52]. Although methyl salicylate was only significantly negatively correlated with frankincense in this experiment, He et al. also showed that methyl salicylate is also a differential substance that affects the aroma of oolong tea, and it may be involved in the formation of floral aromas [53]. In addition, Guo et al. found that 2,4-heptadienal has a sweet and refreshing floral and fruity aroma, and is an important volatile substance that constitutes the characteristic aroma of Shuixian [9]. Although the correlation between 2,4-heptadienal with floral and fruity aromas and the evaluation results was not significant, the OAV value was extremely high and positively correlated with floral and fruity aromas and honeyed aromas. Therefore, 2,4-heptadienal is also a potential aroma substance that should not be ignored. The wide variety and differences in the content of volatile substances due to origin and processing techniques are the basis for the formation of the distinctive aromas of the six rock teas, which give rise to different olfactory experiences.

If the influence of the place of origin is not considered, in order to retain the floral and fruity aromas of the tea as much as possible and reduce the roasted aroma, the processing technique can be tried multiple times. Li et al. found that the aroma score of the Shuixian that was roasted twice was higher than that of the Shuixian that was roasted once, which may be the result of roasting promoting the Maillard reaction and stimulating the conversion of glycoside aroma components [54]. At the same time, the storage of tea should be protected from moisture and aging. Tea that has been stored for too long can be considered for re-roasting to improve the taste and smell.

However, the sample size used in this study was small, and more large-scale experiments are needed to verify the results.

### 4.3. Important Non-Volatile Substances Confer the Health Benefits of Rock Tea

Water-soluble extractives are the sum of the water-soluble substances in tea, including amino acids, soluble sugars, caffeine, soluble proteins, etc. They are not only closely related to the taste and aroma of the tea infusion, but also determine the health benefits of tea [55]. The HT in Tie Luo Han is extremely significantly higher than that in WT, which shows that HT tea infusion has a higher extraction rate, less nutrient loss, more nutrients that can ultimately be absorbed by the human body, and an overall stronger health benefit.

In addition to monosaccharides and disaccharides, soluble sugars also include polysaccharides. Tea polysaccharides account for about 20% of soluble sugars and are a type of complex polysaccharide with certain physiological activities, including hypoglycemic, anticoagulant, and antithrombotic effects [56,57]. The experimental results show that there was no significant difference in the soluble sugars of the six rock teas. Therefore, the health benefits of the six rock teas in terms of hypoglycemic, anticoagulant, and antithrombotic effects may also be similar.

Free amino acids are a group of substances that are mainly responsible for the umami flavor, and are dominated by theanine, glutamic acid, and aspartic acid. Theanine, in particular, has good blood-pressure-lowering, relaxing, and weight-loss effects [58]. The free amino acid content of Wuyi rock tea is significantly higher than that of Huizhou rock tea, so Wuyi rock tea may be better at promoting lower blood pressure, relaxation, and weight loss.

Tea polyphenols are a complex of polyhydroxyphenolic compounds in tea, consisting of more than 30 phenolic substances. The main component is catechin compounds, which have antioxidant and anti-inflammatory effects [59]. Among the Huizhou rock teas, only Shuixian is significantly higher than Wuyi rock tea, while the other two are significantly lower. Different origins or processing techniques give the six rock teas their own strengths, giving them different degrees of antioxidant and anti-inflammatory effects.

We also noticed that Huizhou rock tea has a relatively high content of water leachables and a slightly higher content of soluble sugars, but a slightly lower content of free amino acids. Focusing solely on water leachables does not give a good indication of the specific content of nutrients. This difference between rock teas is a strong reflection of their own health benefits.

## 5. Conclusions

This study used high-performance liquid chromatography (HPLC), gas chromatography–mass spectrometry (GC–MS), and sensory evaluation methods, combined with multivariate statistical analysis, to determine the taste and aroma characteristics of six types of rock tea from the two regions and compare their quality components. The difference in the taste of rock tea mainly comes from the significant difference in the content of theanine (VIP > 1), and theanine and other health-promoting components are closely related to the origin and quality. Among the 23 volatile substances, 10 aroma components were predicted to form the characteristic aroma of rock tea (VIP > 1, OAV ≥ 1). Among them, hexanoic acid, hexyl ester, benzyl nitrile, and methyl salicylate were the most important volatile substances, and 2,4-heptadienal was also predicted to be a potential aroma substance. Combined with a number of experiments, this shows that the processing technology and origin do lead to the difference in the quality of rock tea between Wuyi and Huizhou, which can provide ideas for the subsequent introduction and production of rock tea, as well as a reference for consumers’ choice of rock tea.

## Figures and Tables

**Figure 1 foods-14-00004-f001:**
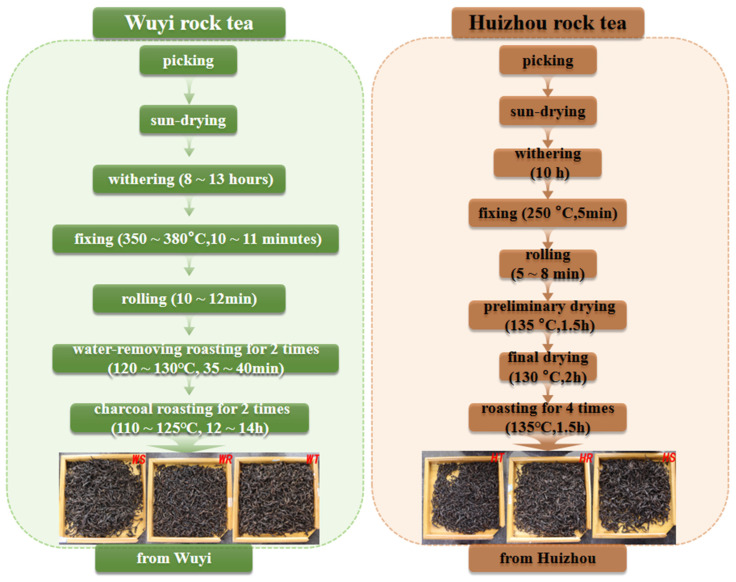
Production process flow chart for 6 kinds of rock tea.

**Figure 2 foods-14-00004-f002:**
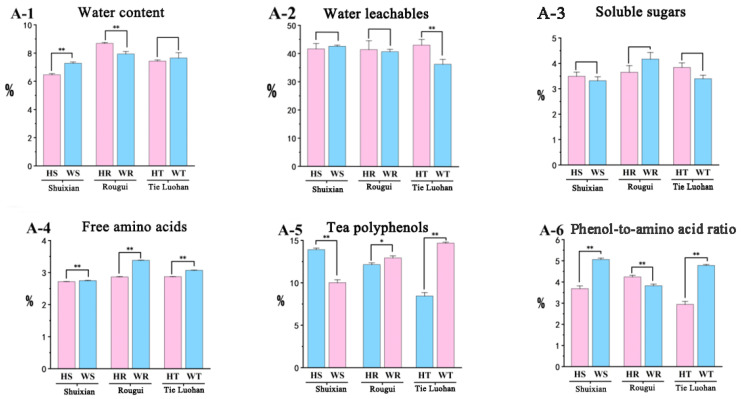
Basic biochemical experiments determined the content of six indicators of six types of rock tea, and the content is expressed as the average value (%) ± standard deviation (*n* = 3). Significant differences are marked according to one-way ANOVA and Tukey’s post hoc test (* *p* < 0.05; ** *p* < 0.01). (**A-1**) Water content of 6 kinds of rock tea. (**A-2**) Water leachables content of 6 kinds of rock tea. (**A-3**) Soluble sugars content of 6 kinds of rock tea. (**A-4**) Free amino acids content of 6 kinds of rock tea. (**A-5**) Tea polyphenols content of 6 kinds of rock tea. (**A-6**) Phenol-to-amino acid ratio of 6 kinds of rock tea.

**Figure 3 foods-14-00004-f003:**
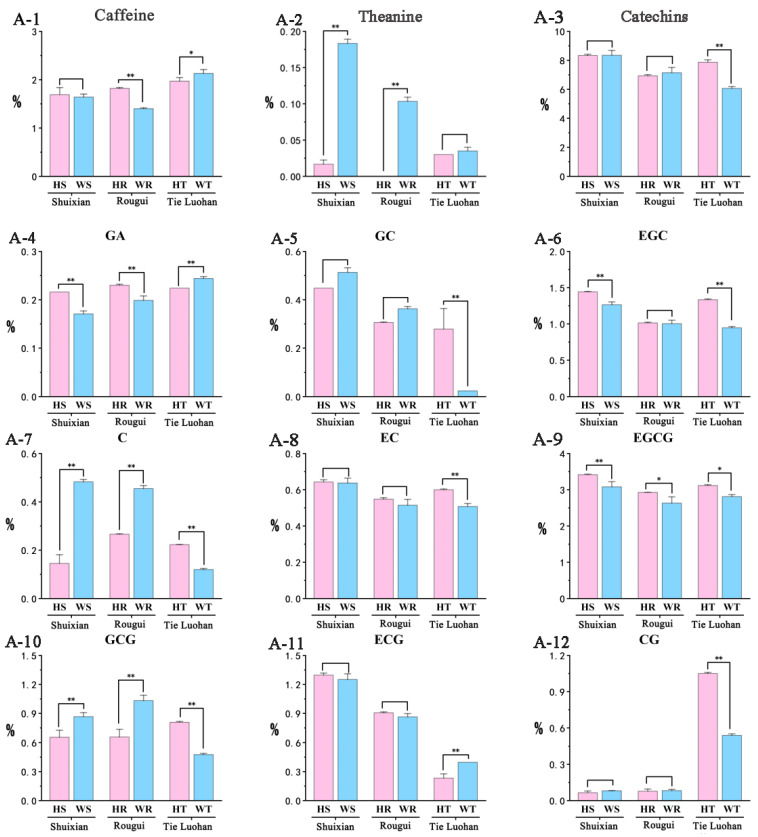
The contents of caffeine, theanine, catechins, and their monomers in the six rock teas were determined using high-performance liquid chromatography (HPLC), and the content is expressed as the mean value (%) ± standard deviation (n = 3). Significant differences are marked according to one-way ANOVA and Tukey’s post hoc test (* *p* < 0.05; ** *p* < 0.01). (**A-1**) Caffeine content of 6 kinds of rock tea. (**A-2**) Theanine content of 6 kinds of rock tea. (**A-3**) Catechins content of 6 kinds of rock tea. (**A-4**) GA content of 6 kinds of rock tea. (**A-5**) GC content of 6 kinds of rock tea. (**A-6**) EGC content of 6 kinds of rock tea. (**A-7**) C content of 6 kinds of rock tea. (**A-8**) EC content of 6 kinds of rock tea. (**A-9**) EGCG content of 6 kinds of rock tea. (**A-10**) GCG content of 6 kinds of rock tea. (**A-11**) ECG content of 6 kinds of rock tea. (**A-12**) CG content of 6 kinds of rock tea.

**Figure 4 foods-14-00004-f004:**
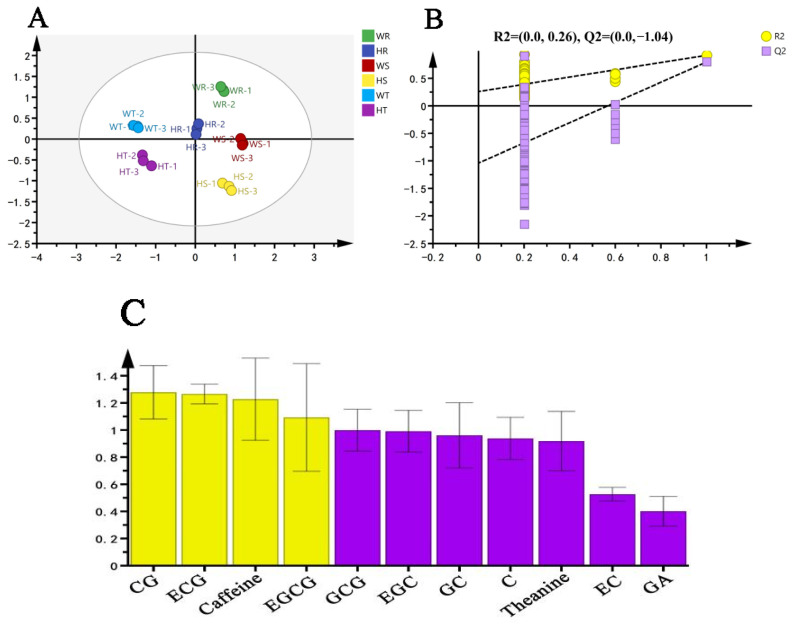
(**A**) OPLS-DA score plot. (**B**) Cross-validation model: 200-fold cross-validation model results: R^2^ = 0.26, Q^2^ = −1.04, indicating that the OPLS-DA discriminant model is not overfitted and the model is relatively reliable. (**C**) VIP score plot; yellow bars represent non-volatile compounds with VIP > 1; purple bars represent non-volatile compounds with VIP < 1.

**Figure 5 foods-14-00004-f005:**
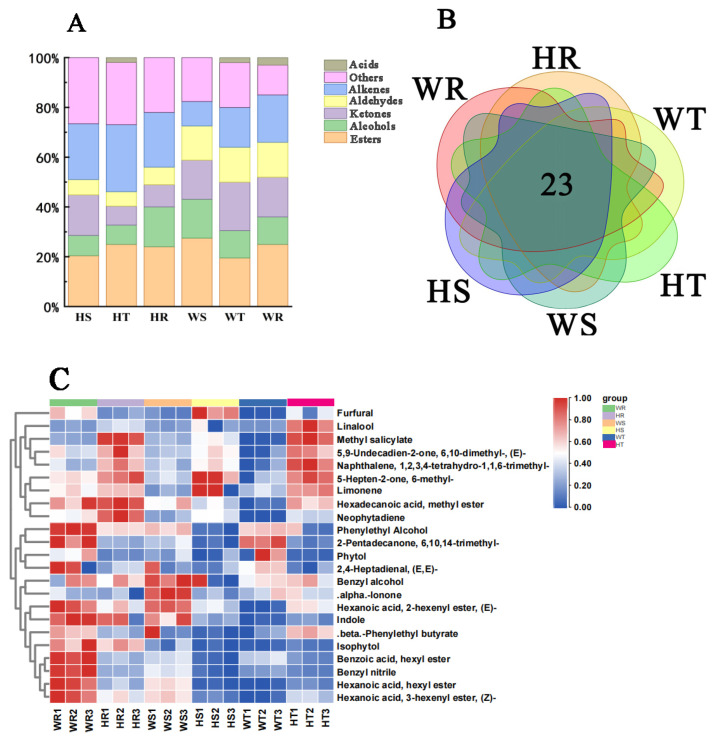
(**A**) Volatile substance stacking chart of 6 rock teas. (**B**) Volatile substance Venn diagram of 6 rock teas. (**C**) Heat map of 23 common volatile substances.

**Figure 6 foods-14-00004-f006:**
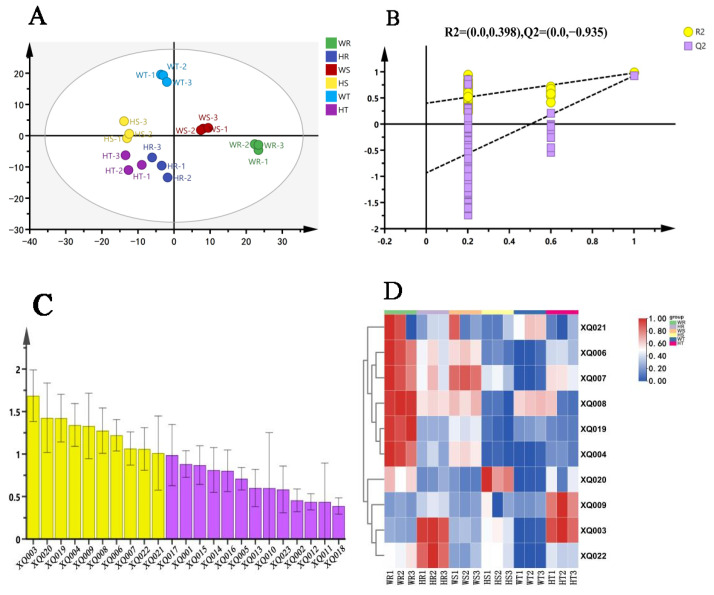
(**A**) OPLS-DA score plot. (**B**) Cross-validation model: 200-fold cross-validation model results: R^2^ = 0.398, Q^2^ = −0.935, indicating that the OPLS-DA discriminant model is not overfitted and the model is relatively reliable. (**C**) VIP score plot, with yellow bars representing volatile compounds with VIP > 1 and purple bars representing volatile compounds with VIP < 1. (**D**) Heat map of the 10 important volatile compounds.

**Figure 7 foods-14-00004-f007:**
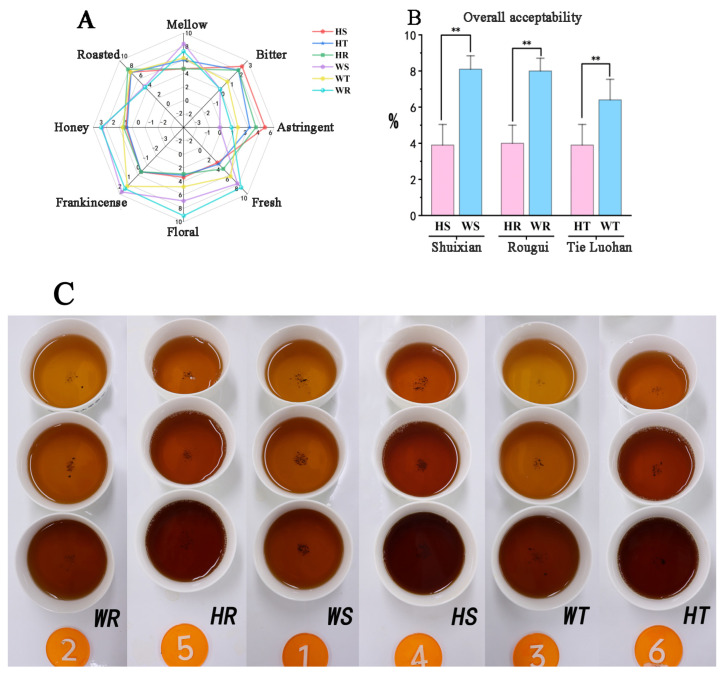
(**A**) Radar chart of the sensory evaluation results of 6 rock teas; (**B**) overall acceptability chart of the sensory evaluation of 6 rock teas (** *p* < 0.01); (**C**) tea infusion chart of 6 rock teas.

**Figure 8 foods-14-00004-f008:**
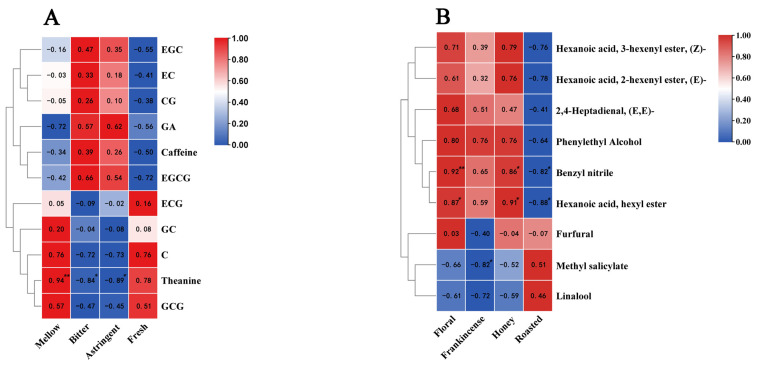
(**A**) Correlation analysis chart of the taste of the six rock teas and key non-volatile substances. (**B**) Correlation analysis chart of the odor of the six rock teas and key volatile substances (* *p* < 0.05; ** *p* < 0.01).

**Table 1 foods-14-00004-t001:** Table of 10 important volatile compound odors.

XQ. No.	Volatile Compound	CAS Number	Odor Type	VIP	1OT (μg/kg)
XQ003	Methyl salicylate	000119-36-8	Peppermint, minty, fresh, sweet	1.69	40 ^a^
XQ020	Furfural	000098-01-1	Sweet, bready, caramel-like	1.43	770 ^b^
XQ019	Benzyl nitrile	000140-29-4	Aromatic odor, pungent	1.42	1 ^c^
XQ004	Hexanoic acid, hexyl ester	006378-65-0	Fruity, green	1.34	16 ^a^
XQ009	Linalool	000078-70-6	Floral, citrus-like aroma	1.33	6 ^a^
XQ008	Phenylethyl Alcohol	000060-12-8	Sweet, floral, rose-like, caramel-like	1.28	45 ^a^
XQ006	Hexanoic acid, 3-hexenyl ester, (Z)-	031501-11-8	Fruity, green	1.22	16 ^a^
XQ007	Hexanoic acid, 2-hexenyl ester, (E)-	053398-86-0	Sweet, fruity	1.07	1.16 ^d^
XQ022	Neophytadiene	000504-96-1	Clear	1.06	n.f.
XQ021	2,4-Heptadienal, (E,E)-	004313-03-5	Fatty, flowery	1.01	0.032 ^e^

Note. n.f.: not found in the literature; 1OT: odor threshold in water; a–e: the thresholds for volatile compounds in water mentioned in the literature are from the following references, where the a–e labels come from the literature [33,34,35,38,39,42].

**Table 2 foods-14-00004-t002:** Table of OAV values for 10 important volatile compounds.

XQ. No.	OAV					
	WR	HR	WS	HS	WT	HT
XQ003	1.53	3.90	1.69	2.34	0.82	3.82
XQ020	0.05	0.01	0.01	0.06	0.00	0.03
XQ019	151.50	65.48	86.60	39.51	59.03	49.53
XQ004	7.61	3.46	5.29	1.75	1.66	2.55
XQ009	5.07	7.21	5.11	7.08	4.85	12.67
XQ008	1.85	1.10	1.21	0.19	1.20	0.18
XQ006	7.55	5.08	5.70	2.99	2.37	4.32
XQ007	51.62	35.47	49.97	30.29	14.16	35.90
XQ022	-	-	-	-	-	-
XQ021	2335.31	863.13	503.44	370.31	1589.69	569.69

-: Indicates that the calculation could not be performed.

## Data Availability

The original contributions presented in the study are included in the article/Appendix A, further inquiries can be directed to the corresponding author.

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
