# Peer review of "Differences in the Quality Components of Wuyi Rock Tea and Huizhou Rock Tea"

_foods, 2024, doi:10.3390/foods14010004_

Round 1
Reviewer 1 Report
Comments and Suggestions for Authors
The manuscript "Differences in the quality components of Wuyi rock tea and Huizhou rock tea" is very interesting. It deals with examining the influence of origin and postharvest processing on the quality of the taste and aroma of tea. My suggestions are the following:
Material and methods: 2.1. Samples
There is less information for Huizhou rock tea than for Wuyi rock tea!
First mising fixing, preliminary and final drying duration. Additionally, missing roasting temperature and duration!
In Figure 1, under each box, write the origin (locality) to make the presentation more complete.
Author Response
Comments 1: [The manuscript "Differences in the quality components of Wuyi rock tea and Huizhou rock tea" is very interesting. It deals with examining the influence of origin and postharvest processing on the quality of the taste and aroma of tea. My suggestions are the following:
Material and methods: 2.1. Samples
There is less information for Huizhou rock tea than for Wuyi rock tea!
First mising fixing, preliminary and final drying duration. Additionally, missing roasting temperature and duration!
In Figure 1, under each box, write the origin (locality) to make the presentation more complete.]
Response 1:[Thank you for pointing this out.We agree with this comment.] Therefore, we have added information about Huizhou rock tea and modified Figure 1.

Reviewer 2 Report
Comments and Suggestions for Authors
Manuscript ID: foods-3359038 “Differences in the quality components of Wuyi rock tea and Huizhou rock tea” has an innovative approach. The manuscript presents text organization, quality figures and an appropriate methodology. The results confirm the described methodology. The discussion is well-founded and contains current references. The manuscript addresses a subject that is relevant to readers of this journal. I suggest that the present manuscript needs minor adjustments .
Abstract: Authors need to add numerical values to corroborate the statements.
Figure 1 should be inserted after the description of the topic "2.1. Samples"
2.7. Sensory evaluation: Please cite the total number of "judges" who participated in the sensory evaluation.
Conclusion: The authors need to describe the importance of the study and how it can have practical and beneficial application for consumers.
Author Response
Comments 1: [Manuscript ID: foods-3359038 “Differences in the quality components of Wuyi rock tea and Huizhou rock tea” has an innovative approach. The manuscript presents text organization, quality figures and an appropriate methodology. The results confirm the described methodology. The discussion is well-founded and contains current references. The manuscript addresses a subject that is relevant to readers of this journal. I suggest that the present manuscript needs minor adjustments .
Abstract: Authors need to add numerical values to corroborate the statements.
Figure 1 should be inserted after the description of the topic "2.1. Samples"
2.7. Sensory evaluation: Please cite the total number of "judges" who participated in the sensory evaluation.
Conclusion: The authors need to describe the importance of the study and how it can have practical and beneficial application for consumers.]
Response 1:[Thank you for pointing this out.We agree with this comment.] Therefore, in the abstract, we have added numerical values to substantiate our statements. We have put Figure 1 in the right place. For “2.7. Sensory evaluation” we cite the total number of “judges” involved in the sensory evaluation. In “Conclusion” we stated the actual usefulness of the study.

Reviewer 3 Report
Comments and Suggestions for Authors
- Page 4, line 158: CAR means Carboxen, not carboxylic acid. Please, correct.
- Page 5, lines 174-175: according to the IUPAC rules, the abbreviation for retention time is tR, not RT. Please, correct.
- p. 3.2.: HPLC chromatogram should be added.
- p. 3.3.: HS SPME GC-MS chromatogram should be added.
Author Response
Comments 1: [- Page 4, line 158: CAR means Carboxen, not carboxylic acid. Please, correct.
- Page 5, lines 174-175: according to the IUPAC rules, the abbreviation for retention time is tR, not RT. Please, correct.
- p. 3.2.: HPLC chromatogram should be added.
- p. 3.3.: HS SPME GC-MS chromatogram should be added.]
Response 1:[Thank you for pointing this out.We agree with this comment.]Therefore, we have therefore corrected “carboxylic acid” to “Carboxen”. We have corrected “RT” to “tR”. HPLC chromatogram and HS SPME GC-MS chromatogram have been added to the attachment.

Round 2
Reviewer 3 Report
Comments and Suggestions for Authors
Accept in present form.